



# SCOPE Climate: a 142-year daily high-resolution ensemble meteorological reconstruction dataset over France

Laurie Caillouet[1,a], Jean-Philippe Vidal[1], Eric Sauquet[1], Benjamin Graff[2], and Jean-Michel Soubeyroux[3]

[1]Irstea, UR RiverLy, Centre de Lyon-Villeurbanne, 5 rue de la Doua CS 20244, 69625 Villeurbanne, France
[2]Compagnie Nationale du Rhône (CNR), 2 rue André Bonin, 69004 Lyon, France
[3]Météo-France, Direction de la Climatologie et des Services Climatiques, 42 avenue Coriolis, 31057 Toulouse Cedex 1, France
[a]now at: INRS, Centre Eau Terre Environnement, 490 rue de la Couronne, Québec (Québec) G1K 9A9, Canada

*Correspondence to:* Laurie Caillouet (laurie.caillouet@gmail.com)

**Abstract.** SCOPE Climate (Spatially COherent Probabilistic Extended Climate dataset) is a 25-member ensemble of 142-year daily high-resolution reconstructions of precipitation, temperature and Penman-Monteith reference evapotranspiration over France, from 1 January 1871 to 29 December 2012. SCOPE Climate provides an ensemble of 25 spatially coherent gridded multivariate time series. It is derived from the statistical downscaling of the Twentieth Century Reanalysis (20CR) by the

SCOPE method (Spatially COherent Probabilistic Extended method) which is based on the analogue approach. SCOPE Climate performs well in comparison to both dependent and independent data for precipitation and temperature. The ensemble aspect corresponds to the uncertainty related to the SCOPE method. SCOPE Climate is the first century-long gridded high-resolution homogeneous dataset available over France and thus paves the way for improving the knowledge on specific past meteorological events or for improving the knowledge on climate variability since the end of the 19th century. This dataset has

also been designed as a forcing dataset for long-term hydrological applications and studies of the hydrological consequences of climate variability over France. SCOPE Climate is freely available for any non-commercial use, and can be downloaded as NetCDF files from http://doi.org/10.5281/zenodo.1299760 for precipitation, http://doi.org/10.5281/zenodo.1299712 for temperature, and http://doi.org/10.5281/zenodo.1251843 for reference evapotranspiration.

## 1   Introduction to SCOPE Climate

Historical surface meteorological observations like precipitation and temperature are more and more scarce and sparse when going back in time before the 1950s, with the number of available stations in databases reduced to only a few in the early 1870s even in a data-rich country like France. Data rescue efforts are on-going (Jourdain et al., 2015) but the present state of databases prevents performing century-long analyses of climate variability and extremes over large regions in a both spatially and temporally homogeneous way. In some other countries with an observation network that started earlier with a higher

density like the UK, long-term gridded datasets of daily rainfall have been derived based on the interpolation between stations from 1890 onwards (Keller et al., 2015). Similarly, daily gridded estimates of potential evapotranspiration have recently been derived from the network of mean monthly temperature observations in the UK from 1891 onwards (Tanguy et al., 2018).





The recent release of two global reanalyses spanning the entire twentieth century – the Twentieth Century Reanalysis (20CR, Compo et al., 2011) and the European Reanalysis of the Twentieth Century (ERA-20C, Poli et al., 2016) – provides the opportunity to reconstruct long-term local-scale meteorological data without having to rely only on a ground observation network that is loose back in time and non-homogeneous in both space and time. Indeed, statistical downscaling methods may

be used to derive daily local-scale near-surface meteorological variables from synoptic-scale atmospheric variables provided by such global extended reanalyses. 20CR has already been downscaled for specific regions in France (e.g. Kuentz et al., 2015), or for the entire country (Minvielle et al., 2015; Dayon et al., 2015; Bonnet et al., 2017) but without a locally optimized method and/or in a deterministic way, thus preventing any detailed local analyses that would take account of the downscaling uncertainty.

Caillouet et al. (2016, 2017) proposed to statistically downscale 20CR with an ensemble method optimized locally over France – based on an analogue resampling of the existing 50-year long Safran near-surface reanalysis (Vidal et al., 2010) – in order to improve the knowledge on past hydrometeorological conditions and climate variability since the end of the 19th century in France. SCOPE Climate (Spatially COherent Probabilistic Extended Climate dataset), one of the resulting datasets, is a daily high-resolution ensemble reconstruction of precipitation, temperature and Penman-Monteith reference evapotranspiration

fields in France from 1 January 1871 to 29 December 2012. Available on a 8-km grid, SCOPE Climate has the appropriate space and time resolutions for hydrological applications (see Caillouet et al., 2017, for an application on reconstructing historical low-flow events). This paper proposes to make SCOPE Climate accessible to the research community.

The most general choices have been made in the creation of SCOPE Climate to promote the widest possible use of the dataset. The length (142 years), the spatial availability (entire France) as well as the ensemble aspect (25 members) will enable

various spatio-temporal analyses to enhance the knowledge of past meteorology and climate, and their hydrological impacts. SCOPE Climate moreover provides homogeneous time series that will ensure the spatial consistency required for all studies.

This paper presents the SCOPE Climate dataset, first by introducing in Sect. 2 the methodological framework developed to derive it. Characteristics of SCOPE Climate are then detailed in Sect. 3 through examples and validation results. Section 4 describes how to access the dataset, and Sect. 5 considers some of its limitations.

## 25  2   Statistical downscaling of 20CR with the SCOPE method

### 2.1   Data

SCOPE Climate has been derived based on several datasets presented below.

### 2.1.1   Source for local predictands: Safran

Safran is the French near-surface reanalysis available at the hourly time scale and 8-km spatial resolution, from the 1 Au-

gust 1958 onwards (Vidal et al., 2010). Safran has been derived based on an optimal interpolation between all surface observations available in the Météo-France database and a first guess from the ERA-40 reanalysis (Uppala et al., 2005) for 608





climatically homogeneous zones paving France (Quintana-Seguí et al., 2008). Daily reference evapotranspiration has been computed based on the hourly Penman-Monteith formula (Allen et al., 1998).

Local predictands used to derive SCOPE Climate are gridded daily precipitation, temperature, and reference evapotranspiration over the period 1 August 1958 to 31 July 2008.

### 2.1.2 Source for atmospheric predictors: 20CR

The Twentieth Century Reanalysis (20CR) is a large-scale reanalysis spanning the entire twentieth century and developed by the National Oceanic and Atmospheric Administration (NOAA) (Compo et al., 2011). 20CR V2 chosen here only uses 6-hourly surface level pressure (SLP) observations from the International Surface Pressure Databank (ISPD v2.2, Compo et al., 2010) in the Ensemble Kalman Filter assimilation process. This choice has been made to avoid inhomogeneities due to the assimilation of observations from different systems.

Six atmospheric predictors are considered in the downscaling process used to derive SCOPE Climate: temperature at 925 hPa and 600 hPa, geopotential height at 1000 hPa and 500 hPa, vertical velocity at 850 hPa, precipitable water content, relative humidity at 850 hPa – all from the six-hourly analysis –, and large-scale 2m temperature (T2m) from the six-hourly forecast.

20CR predictors are available at 2.0° spatial resolution and 6-hourly temporal resolution from 1 January 1871 to 31 December 2012. Five predictors – temperature, geopotential height, vertical velocity, precipitable water content, and relative humidity – have been spatially interpolated on the 2.5° grid required in the first step of the downscaling method. The ensemble mean of the 56-member ensemble of 20CR has been considered for building SCOPE Climate (see Sect. 5).

### 2.1.3 Source for oceanic predictor: ERSST

The NOAA Extended Reanalysis Sea Surface Temperature (ERSST) version 3b is a global sea surface temperature (SST) reanalysis, available at a 2.0° spatial resolution and a monthly temporal resolution since 1 January 1854 (Smith and Reynolds, 2003; Smith et al., 2008). Like 20CR, this version does not use satellite data to avoid inhomogeneities. Monthly values of SST over the optimised grid cell south of Brittany (4° W, 46° N) over the period 1871-2012 were extracted and interpolated with splines to the daily time scale required by the downscaling process (see Appendix B in Caillouet et al., 2017), and constitutes the 7th large-scale predictor of the SCOPE method.

### 2.2 The SCOPE statistical downscaling method

The Spatially COherent Probabilistic Extension method (SCOPE, Caillouet et al., 2017) is the statistical method used to downscale 20CR and derive the SCOPE Climate dataset. SCOPE is an extension of the Statistical ANalogue Downscaling method for HYdrology (SANDHY, Ben Daoud et al., 2011, 2016; Radanovics et al., 2013). It can be divided in four steps that have been briefly described in the Appendix B of Caillouet et al. (2017), and that are detailed in the paragraphs below:

– Applying the SANDHY method (Sect. 2.2.1),

  – Subselecting SANDHY analogues for reconstructing both precipitation and temperature (Sect. 2.2.2),



- Correcting for precipitation bias (Sect. 2.2.3),

- Ensuring spatial coherence (Sect. 2.2.4).

### 2.2.1 Step 1: Applying the SANDHY method

The SANDHY method is an ensemble statistical downscaling method following an analogue approach. It is based on the
idea introduced by Lorenz (1969) that similar atmospheric situations lead to similar local effects. The analogue approach
uses two concurrent datasets, a large-scale reanalysis containing predictors and a local-scale meteorological dataset containing
predictands. Both datasets should be available on an archive period where the predictors-predictand relationship is set up. In a
reconstruction set-up, the large-scale reanalysis is then the only dataset that must be available over the period to reconstruct,
called the target period. Large-scale predictors from any date in the target period are compared to those of the archive period.
Dates from the archive period with the most similar predictors are then chosen as analogues. Local-scale predictands from the
selected analogue dates are taken as an ensemble of plausible predictand values for the target date.

SANDHY predictand is daily precipitation, as this method was initially aimed at quantitative precipitation forecasting. The
predictors are used in four analogy levels optimized by Ben Daoud et al. (2011, 2016). The first level selects N analogue days
on temperature at 925 hPa and 600 hPa with the exclusion of a 4-day window around the target date. N is taken as 100 times
the number of years in the archive period. The second level selects 170 analogues on geopotential height at 500 hPa and 1000
hPa. The third level selects 70 analogues thanks to an analogy on vertical velocity at 850 hPa and the final level selects 25
analogues on humidity, considered as the product of the precipitable water content and the relative humidity at 850 hPa.

The similarity criterion used for the analogy levels on temperature, vertical velocity and humidity is the Euclidean distance,
with equal weights when different pressure levels are used. The analogy on geopotential height is measured through the shape
similarity between fields with the Teweles and Wobus (1954) criterion. The predictor domain (spatial domain where the analogy
is sought) depends on the Safran zone considered. The spatial domain for the first, third and fourth analogy levels is the closest
large-scale grid point to each zone. For the analogy level on geopotential, this domain has been optimized by Radanovics
et al. (2013) on the 608 climatically homogeneous Safran zones covering France. Five near-optimal domains are obtained for
each zone in France using an algorithm of growing rectangular domains. The performance criterion for this optimization was
the Continuous Ranked Probability Score (CRPS, Brown, 1974; Matheson and Winkler, 1976), widely used for probabilistic
verification forecast. Domains found from neighbouring zones were also considered if they provide a better performance.

The synthetic diagram in Fig. 1 summarizes the different steps described above. 20CR variables are used as predictors (see
Sect. 2.1.2) whereas Safran precipitation is selected as predictand (see Sect. 2.1.1). The predictor domains per climatically
homogeneous zones have been optimized with 20CR data using the 1 August 1982-31 July 2002 period, following Radanovics
et al. (2013) method. The archive period over which the predictors-predictand relationship is set up is 1 August 1958 to
31 July 2008. The target period considered is the whole period spanned by 20CR V2 data, i.e. 1 January 1871 to 31 Decem-
ber 2012. The different downscaling steps are applied 5 times using one of the near-optimal analogy domain for geopotential



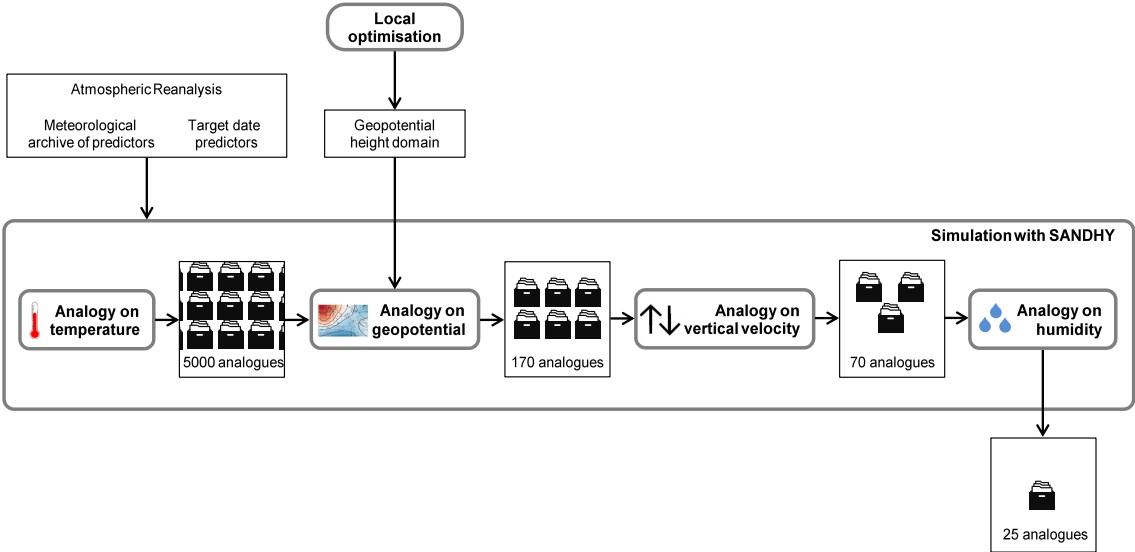

**Figure 1.** Synthetic diagram showing the sequence of analogy steps in the SANDHY method to reconstruct precipitation and temperature, over a given climatically homogeneous zone, and for a specific target date.

at each run. The application of SANDHY with these settings provides an ensemble of 125 analogue dates for each date in the target period, independently for the 608 climatically homogeneous zones covering France.

Analogues dates included in the period 1 August 1958 to 31 July 2008 are then converted to meteorological variables by resampling Safran reanalysis data. The analogues obtained from SANDHY are not only used for precipitation, but also for
temperature and reference evapotranspiration.

### 2.2.2 Step 2: Subselecting SANDHY analogues

The stepwise subselection has been developed by Caillouet et al. (2016) to improve the use of SANDHY in a reconstruction context and for other meteorological variables than precipitation. Indeed, an over-estimation of precipitation in spring and an under-estimation of precipitation in autumn was found in SANDHY outputs for zones with a high seasonal asymmetry (e.g.,
Mediterranean areas). Moreover, winter and summer temperatures were respectively over- and under-estimated. This last result was not unexpected since SANDHY predictors were chosen for their strong relation to precipitation, and not to temperature.

The stepwise subselection is a post-processing approach keeping unchanged the structure of the SANDHY method while reducing the above-mentioned biases. It follows the SANDHY overarching principle, with two new analogy levels applied on the 125 analogues dates resulting from the application of SANDHY. The first level selects 80 analogues out of 125 on the
similarity of SST values. The second level selects 25 analogues on the similarity of T2m values. The number of analogues at each level has been optimized by Caillouet et al. (2016). The similarity criterion for both levels is the Euclidean distance. For





the sake of consistency over France and parsimony of parameters, a single grid point common to all climatically homogeneous zones in France is used for computing the analogy on SST (optimised at location 4° W, 46° N). The grid point for computing the T2m analogy is chosen as the land grid point closest to each climatically homogeneous zone, following the approach used for levels 1, 3 and 4 in the standard SANDHY method. A detailed validation of results from the chain SANDHY + stepwise

(also called SANDHY-SUB) for temperature and precipitation over France has been performed by Caillouet et al. (2016).

### 2.2.3   Step 3: Correcting for precipitation bias

A third step consists in a bias correction of SANDHY-SUB precipitation. Indeed, the median of annual precipitation between Safran and reconstructed precipitation showed a dry bias of around 10% (see Caillouet et al., 2016). Instead of applying common bias correction techniques, a correction approach similar to the one adopted by Sippel et al. (2016) has been considered

here, keeping a maximum inter-variable coherence, inherent to analogue methods.

For each target date between 1871 and 2012, the N analogues giving the lowest precipitation are removed. N analogues are then randomly resampled among the (25 - N) left to keep a 25-member sample size. N is independently defined for each of the 608 zones in France. It is chosen between 0 and 3, so that the bias with respect to Safran data (over the archive period) is minimized. By construction, this number increases with precipitation under-estimation. Importantly, this resampling based

correction of precipitation does not affect the temperature bias and interannual correlation described in Caillouet et al. (2016). This technique allows one to retrieve a near-zero bias in mean interannual precipitation over France.

### 2.2.4   Step 4: Ensuring spatial coherence

The last step to build SCOPE Climate is to add some spatial coherence to ensemble member fields. Indeed, in order to build gridded time series, analogue dates must be combined from one zone to another. The resulting lack of spatial coherence

when using a random combination from one zone to another could be heavily detrimental to the use of SCOPE Climate for applications requiring a spatial coherence, like hydrological studies. This issue is addressed here by applying the Schaake Shuffle procedure, initially developed to reconstruct space-time variability in forecast meteorological fields (Clark et al., 2004). In this approach, the ensemble members are reordered so that their rank correlations across both space and variables match the ones from a randomly picked sample of observed multivariate fields. In the present application, rank correlations are

considered across the 608 climatically homogeneous zones and across the three variables (precipitation, temperature, and reference evapotranspiration). Observed fields are taken from the Safran reanalysis.

For each target date, 25 dates are randomly selected within a 120-day window around the corresponding Julian day and from the period 1 August 1958 to 31 July 2008, a period consistent with the archive period for analogue dates in the SANDHY downscaling step. Observed rank correlations are derived from the Safran multivariate meteorological fields for this ensemble

of 25 dates and applied to the reconstructed ensemble, thus ensuring a spatial and inter-variable coherence of any single ensemble member. As the set of analogues remains the same for each day and is only shuffled across ensemble members, local characteristics such as median ensemble bias do not change.

Figure 2 provides a summary of the different steps of the SCOPE method.



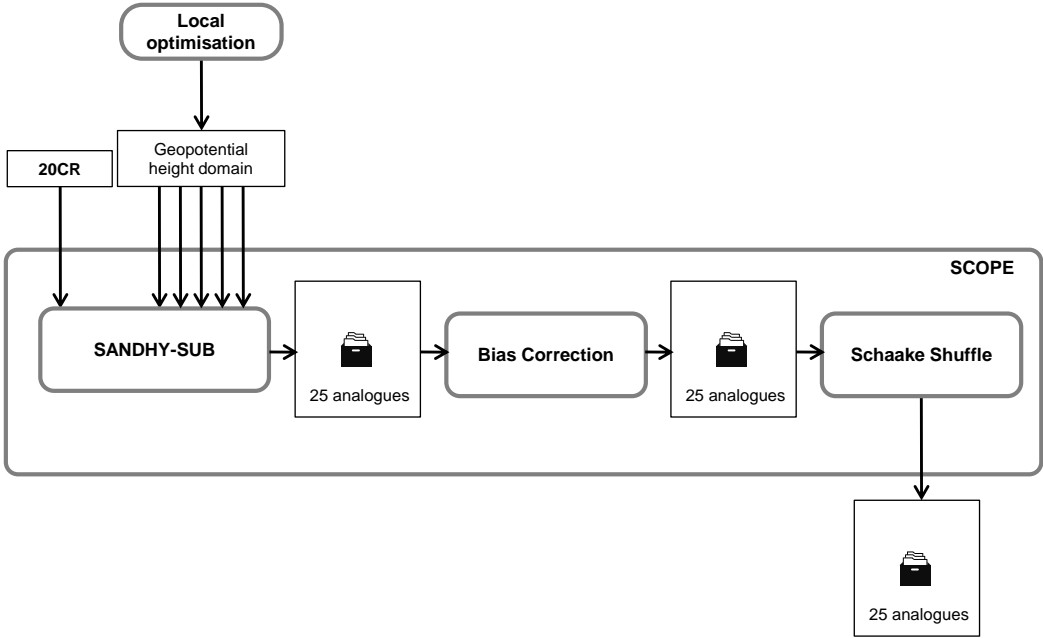

**Figure 2.** Synthetic diagram showing the sequence of steps for the SCOPE method and its use for SCOPE Climate reconstructions.

## 3 The resulting dataset: SCOPE Climate

SCOPE Climate is the gridded climate dataset derived from the downscaling of 20CR by the SCOPE method. It consists in an ensemble of 25 equally plausible individual members of multivariate gridded meterological series. Each member gathers daily gridded times series of precipitation, temperature and reference evapotranspiration for the period 1 January 1871 to
5   29 December 2012 on a 8-km grid over France. For a given date and a given member, values are coherent over space and across the three variables.

The sections below provide weather and climate examples from SCOPE Climate, followed by a validation against Safran.

### 3.1 Weather and climate examples from SCOPE Climate

#### 3.1.1 Daily spatial features: 18-20 January 1910 precipitation and temperature over France

10   January 1910 was a particularly wet month in France, especially in the Seine river catchment. The combination of heavy rain during this month, saturated soils at the end of December 1909, frozen grounds and melting snow, have led to widespread floods in the region (Lang et al., 2013). The succession of several perturbations between the 17th and 20th of January have resulted in above-average precipitation accumulation with more than 130 mm in the Morvan, usually corresponding to the




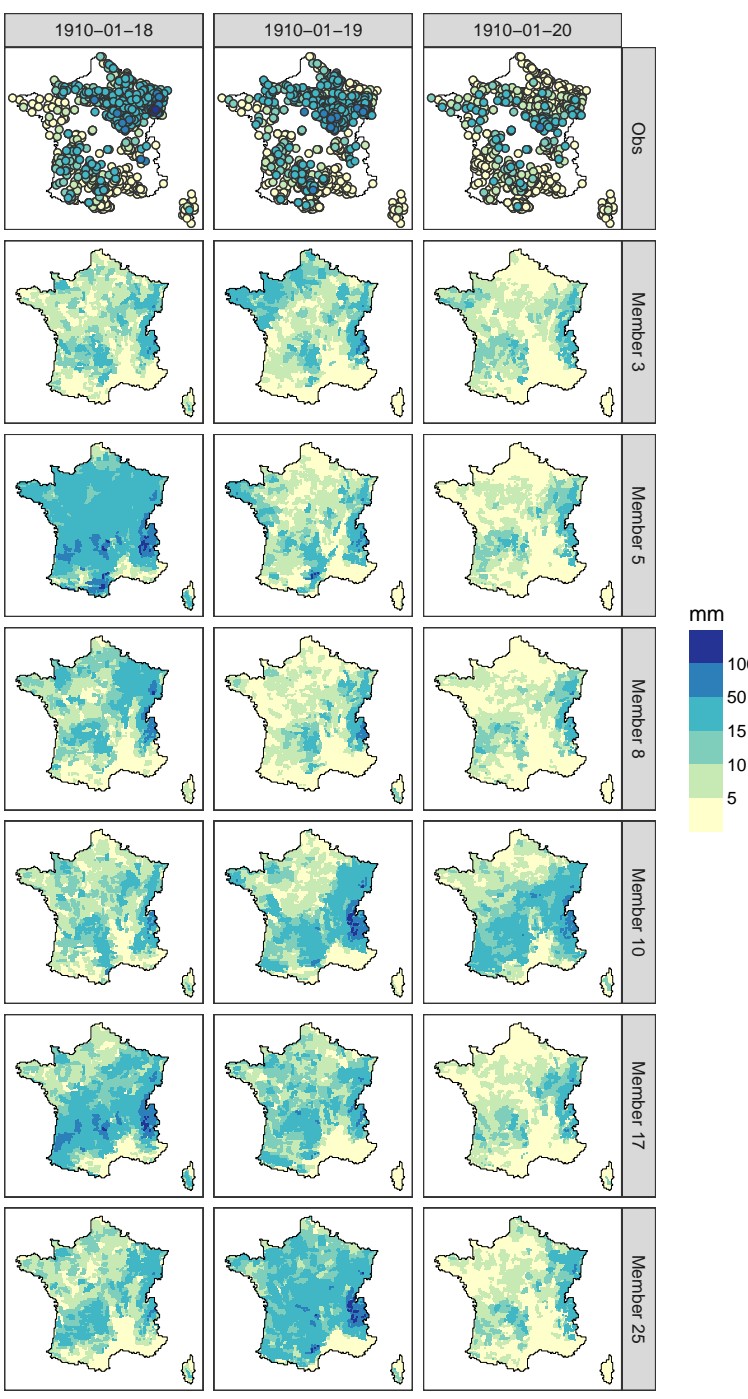

**Figure 3.** 18 to 20 January 1910 precipitation. Top: observations currently available from the Météo-France database. Rows 2-7: 6 randomly selected members of SCOPE Climate.





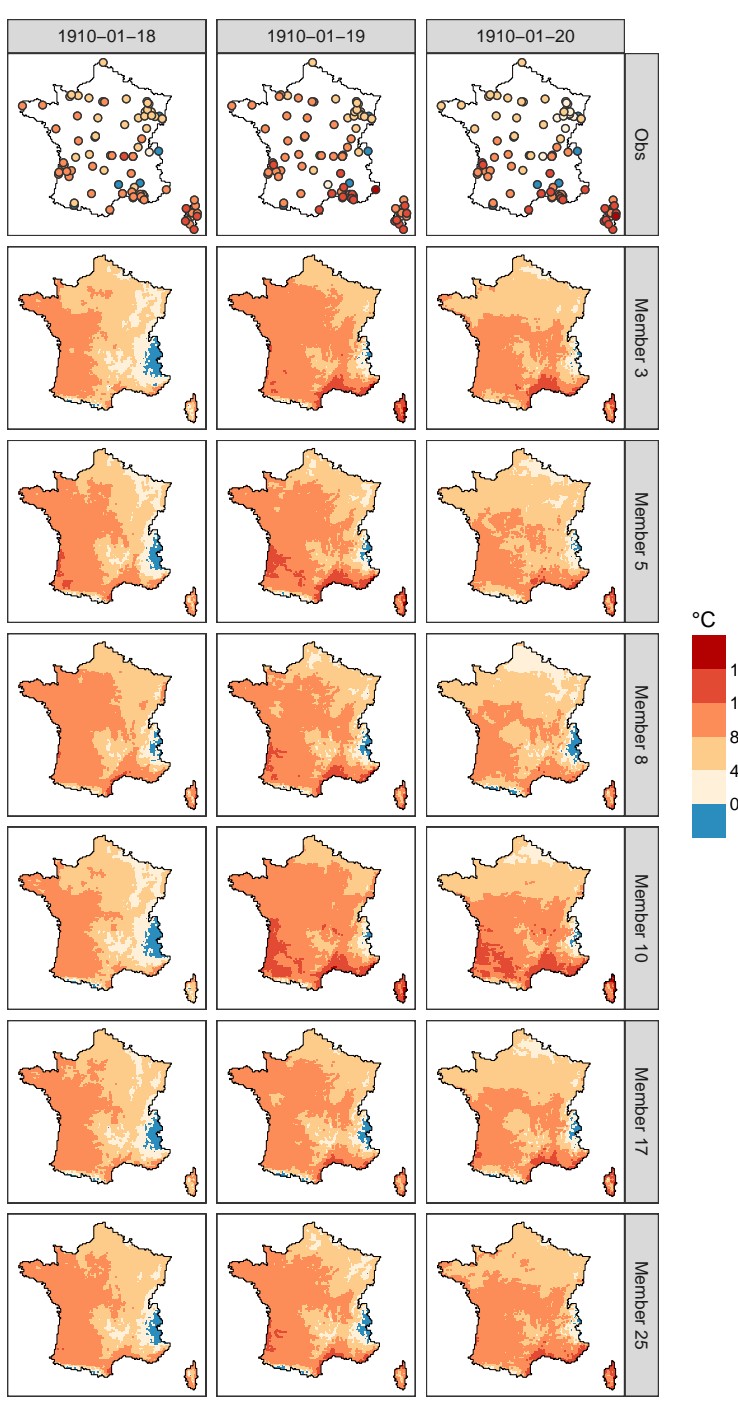

**Figure 4.** As for Fig. 3, but for mean temperature.



amount of rain for the entire month of January[1] (Schneider, 1997). The 18th to 20th of January, corresponding to peak of the event, have been chosen as the first reconstruction example. They are partly responsible for the most studied 100-year flood of the Seine river in Paris and its tributaries (Nouailhac-Pioch and Maillet, 1910; Marti and Lepelletier, 1997; Delserieys and Blanchard, 2014), but also for other important floods on the Rhine (Martin et al., 2011) or Rhône (Pardé, 1925) tributaries.

5     The top row of Fig. 3 shows the maps of 18, 19, 20 January 1910 daily precipitation from available observations in the Météo-France database. It has to be noted that particular efforts of data rescue have been employed for the Seine river basin, resulting in a high density of available observations. From the observations, heavy amounts have been recorded over several mountain ranges, (Morvan, southern Vosges, Jura and northern French Alps), specially for the 18th and 19th of January. The north-west of France (Brittany) and the south-west of France are also affected by high precipitation amounts, respectively on 10  the 20 January and the 18-19 January. The south-east of France and Corsica remained quite dry (except for one station in the mountain range of Corsica). However, several regions suffer from a lack of observations: Picardie (north), the main part of the Loire basin (centre), Jura mountain range (east), most of the Alps, and Provence (south-east).

    Concurrent SCOPE Climate precipitation reconstructions are presented on the rows 2-7 of Fig. 3 with 6 randomly selected members. Some members provide high amounts of precipitation for almost the entire France (on the 18th for member 5 and 15  on the 19th for member 25) with an emphasis on mountain ranges (Vosges, Jura, French Alps). Some members provide high precipitation amounts on a diagonal from the north-east to the south-west (on the 19th and 20th for member 10 and on the 18th for member 17). Others provide high precipitation for particular zones in France (Brittany for member 3 on th 19th, east for member 8 on the 8th). Four members out of the 6 randomly selected members provide more than 100 mm of precipitation for the French Alps and Jura on a particular day (members 5, 10, 17 and 25), a zone without available precipitation observations 20  but where important floods have been recorded (Boudou et al., 2016).

    Figure 4 provides the maps of surface temperature for the 18 to 20 January 1910 from available observations (top) and the same 6 selected members of SCOPE Climate. Spatial patterns of reconstructed temperature are in good agreement between each other and with observations. Values are largely positive over the Seine catchment, in agreement with snow melt records. This example demonstrates the ability of SCOPE Climate to fill in the spatial gaps of missing data and to provide a spatially 25  coherent reconstruction of temperature.

### 3.1.2   Temporal features: time series at Lyon-Bron airport

The evaluation of SCOPE Climate against independent data is possible using homogenized series provided by Météo-France (Moisselin et al., 2002; Moisselin and Schneider, 2002). These series have been computed at a monthly time step using a statistical procedure detecting breaks and outliers in long time series of observations with sufficient quality. The 323 monthly 30  precipitation series and 65 monthly time series of minimum and maximum temperature spanning the whole twentieth century have been retained for the evaluation. Monthly mean temperature time series have been obtained by averaging minimum and maximum temperature as done by Moisselin and Schneider (2002). The closest 8-km grid point to the observation station is selected for the comparison.

---

[1]http://www.meteofrance.fr/actualites/34465980-crue-de-la-seine-quelle-meteo-en-1910, last access 02-19-2018.



**Figure 5.** Lyon-Bron precipitation homogenized time series, corresponding Safran data, and reconstructed series from SCOPE Climate at the annual and seasonal timescales over the 1871–2012 time period. Light and dark purple ribbons define the range and the interquartile range, respectively, of values from SCOPE Climate members. Note the different scales for the y axes.





**Figure 6.** As for Fig. 5, but for temperature.





The Lyon-Bron airport station holds very long time series (since 1881 for precipitation and 1885 for temperature), already used in climate variability studies (see e.g. Thibert et al., 2013). It has been chosen here to compare the annual and seasonal evolution of precipitation (Fig. 5) and temperature (Fig. 6) from (1) the homogenized series, (2) Safran data, and (3) the range of reconstructed series from SCOPE Climate.

Precipitation reconstructions from Fig. 5 show a very satisfactorily reconstructed interannual variability at the annual and seasonal timescales. Except for some specific years, observations (from both homogenized series and Safran) stay inside the range of the 25 reconstructions. At the annual timescale, exceptional years – such as the 1921 dry year (Duband et al., 2004) or the 1960 wet year (Pardé, 1961) – are well captured by SCOPE Climate. The few years before 1897 show a lower reconstruction skill. SCOPE Climate shows a high annual precipitation in 1872, where no data is available. Nevertheless, Gautier et al. (2004)
mentioned one of the biggest flood on the Loire basin in October 1872 and heavy precipitation in Roanne and Macon, two towns close to Lyon. SCOPE Climate has difficulties in simulating the extreme wet spring and dry summer of 1983 (Blanchet, 1984). Nevertheless, other extreme events like the wet winter of 1936 (Pardé, 1937) are well simulated.

Temperature reconstructions from Fig. 6 are compared between the different datasets at the annual and seasonal timescales. The interannual variability is quite satisfactorily simulated, but the uncertainty in reconstructions seems underestimated, with
observations being too often out of the range of SCOPE Climate members. Summer temperatures are generally underestimated whereas winter temperatures are generally overestimated, leading to a systematic under/over-estimation of temperature in hot/cold years. However, the recent trend in spring and summer temperature is captured – although underestimated – by the reconstruction method.

Some hot seasons are well captured, such as the 1942-1948 period for spring, and specially the hot spring of 1945 (Martin,
1946) or the more recent hot summer of 1976 (Brochet, 1977). The cold autumn of 1912 is also well captured by SCOPE Climate (Puiseux, 1913, p. 63). Temperature for other events like the extremely cold winter 1962-1963 (Geneslay, 1964) or the recent hot summer of 2003 (Trigo et al., 2005) are however over/under-estimated respectively (about +2°C for 1963 and -2.3°C for 2003 in comparison to Safran), but still captured.

### 3.2 Performance of SCOPE Climate against reference datasets

The performance of SCOPE Climate is assessed using Safran as reference dataset. First, direct comparisons to Safran are made through the use of inter-annual regimes and visualization of time series at a daily time step in a specific year. Then, different skill scores are used to complete the validation of SCOPE Climate.

### 3.2.1 Performance against the Safran reanalysis

The inter-annual regimes from Safran and SCOPE Climate precipitation, temperature and reference evapotranspiration are
compared to each other on four different case study cells (Fig. 7). Precipitation regimes are well reconstructed by SCOPE Climate. Spring precipitation is slightly over-estimated whereas autumnal precipitation can be slightly under-estimated (for Finistère and Corrèze). These results show that the SCOPE method is able to reproduce strong seasonal cycles (asymmetry spring/autumn) as well as continental regimes. For temperature, Safran regime is always inside the thin range of SCOPE Cli-



**Figure 7.** Top: topographical map of France with location of the 4 case study cells: Finistère in Brittany, Haute-Savoie in the French Alps, Corrèze in the south-west and Cévennes in the Massif Central mountain range. Brown corresponds to altitudes above 650m (up to 2933m). Green corresponds to altitudes under 500m. Bottom: precipitation, temperature and reference evapotranspiration interannual regimes between 1959 and 2007 from Safran (black) and SCOPE Climate (colors), through the 25 individual members, for the case study cells.



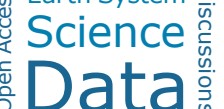

**Figure 8.** Precipitation, temperature and reference evapotranspiration reconstructed series from SCOPE Climate compared to Safran on the 2011 year at a daily time step. Light gray corresponds to the 25 members of SCOPE Climate, Member 1 is plotted in black as an example.

mate regimes in spring and autumn, for all cells. Temperatures are slightly over-estimated in winter and under-estimated in summer. The regimes of reference evapotranspiration show differences between Safran and SCOPE Climate. For all cells, evapotranspiration is under-estimated in spring and summer as well as over-estimated in autumn and winter. Moreover, for the Finistère, SCOPE Climate maximum evapotranspiration is not seen in July, as for Safran, but in August, leading to a shifted sea-

5 sonal cycle. This shows the importance of other variables than temperature in the computation of reference evapotranspiration, and some lack of information on these variables in SCOPE predictors.

The 2011 year has been chosen as an example year to show SCOPE Climate and Safran at a daily time step. This year remains outside of the Safran archive period used to create SCOPE Climate (1958-2008) and is among the hottest years since




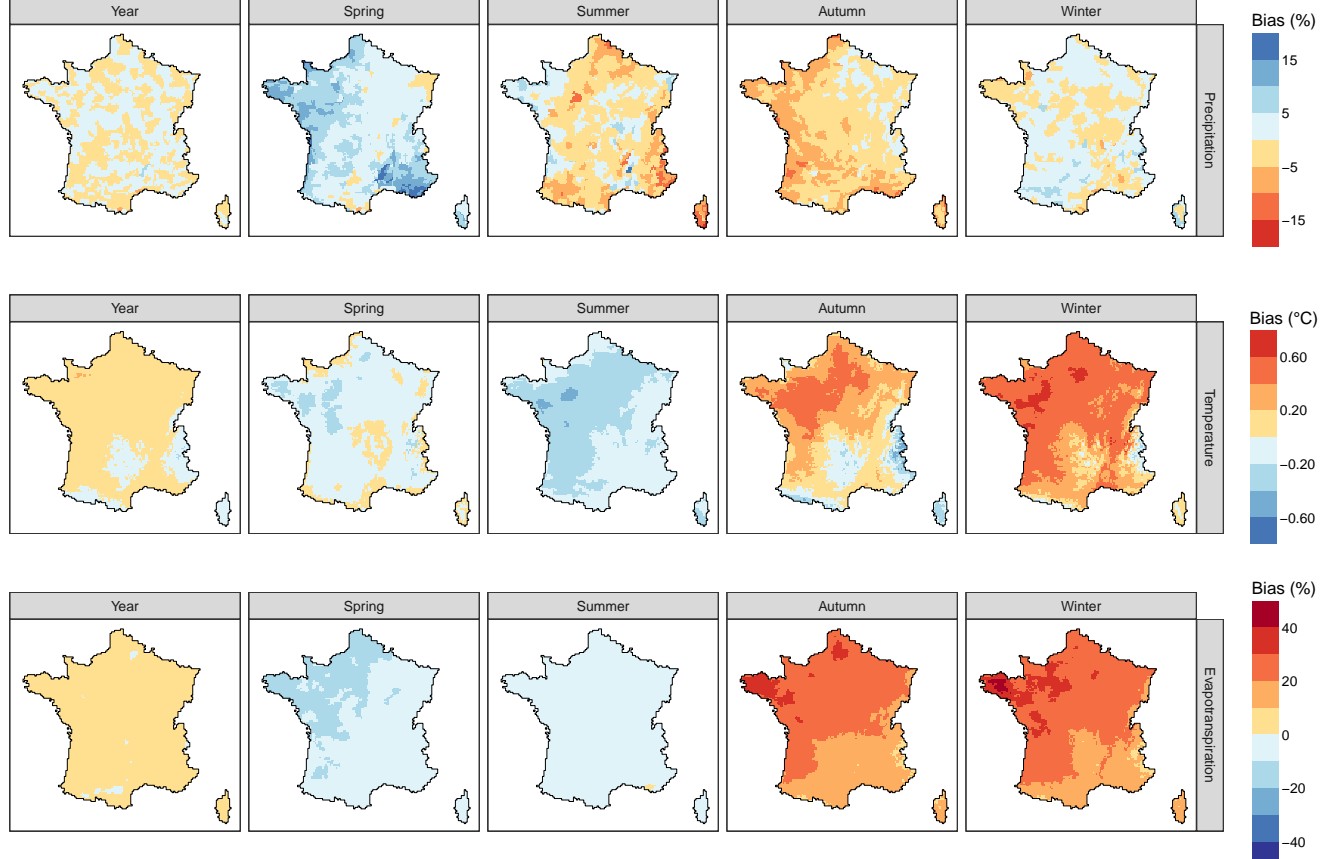

**Figure 9.** Median of annual and seasonal precipitation (top row), temperature (middle row) and evapotranspiration (bottom row) bias between Safran and SCOPE Climate for the 1959–2007 period. Red corresponds to an over-estimation of the reconstructed temperature and reference evapotranspiration as well as an under-estimation of the reconstructed precipitation.

1900[2]. Precipitation, temperature and evapotranspiration time series from both Safran and SCOPE Climate are presented in Fig. 8 for a grid cell included in the Finistère zone (see Fig. 7). Safran precipitation is most of the time included in the range of the 25 reconstructions, except for extremely wet days above 20 mm. There are few days when Safran and all members reconstruct zero precipitation at the same time. Nevertheless, the sequence of alternating dry and wet periods is well respected

5 in SCOPE Climate. Member 1 provides a good reconstruction of Safran precipitation with a slight over-estimation for moderate episodes and an under-estimation for strong episodes. Safran temperature is almost always included in the – sometimes very thin – range of SCOPE Climate temperatures, showing the small bias of the SCOPE method. The overall Safran variability is well reconstructed by Member 1 of SCOPE Climate. The good results obtained for precipitation and temperature do not apply to evapotranspiration. Indeed, even if Safran evapotranspiration is included in the extremely large range of SCOPE Climate

---

[2]http://www.meteofrance.fr/climat-passe-et-futur/bilans-climatiques/autres-annees/bilan-de-lannee-2011, last access 02-20-2018.





reconstructions, the day-to-day variability reconstructed by Member 1 is far too high in comparison to Safran, specially in winter. Moreover, the sign of these day-to-day variations (increase/decrease) is sometimes not the same considering Safran and Member 1 of SCOPE Climate. In spite of a good reconstruction of precipitation and temperature, evapotranspiration results lack of autocorrelation. Caution is therefore required if SCOPE Climate is used for studying specific events where

5    evapotranspiration plays an important role, such as the 2003 drought (Teuling et al., 2013).

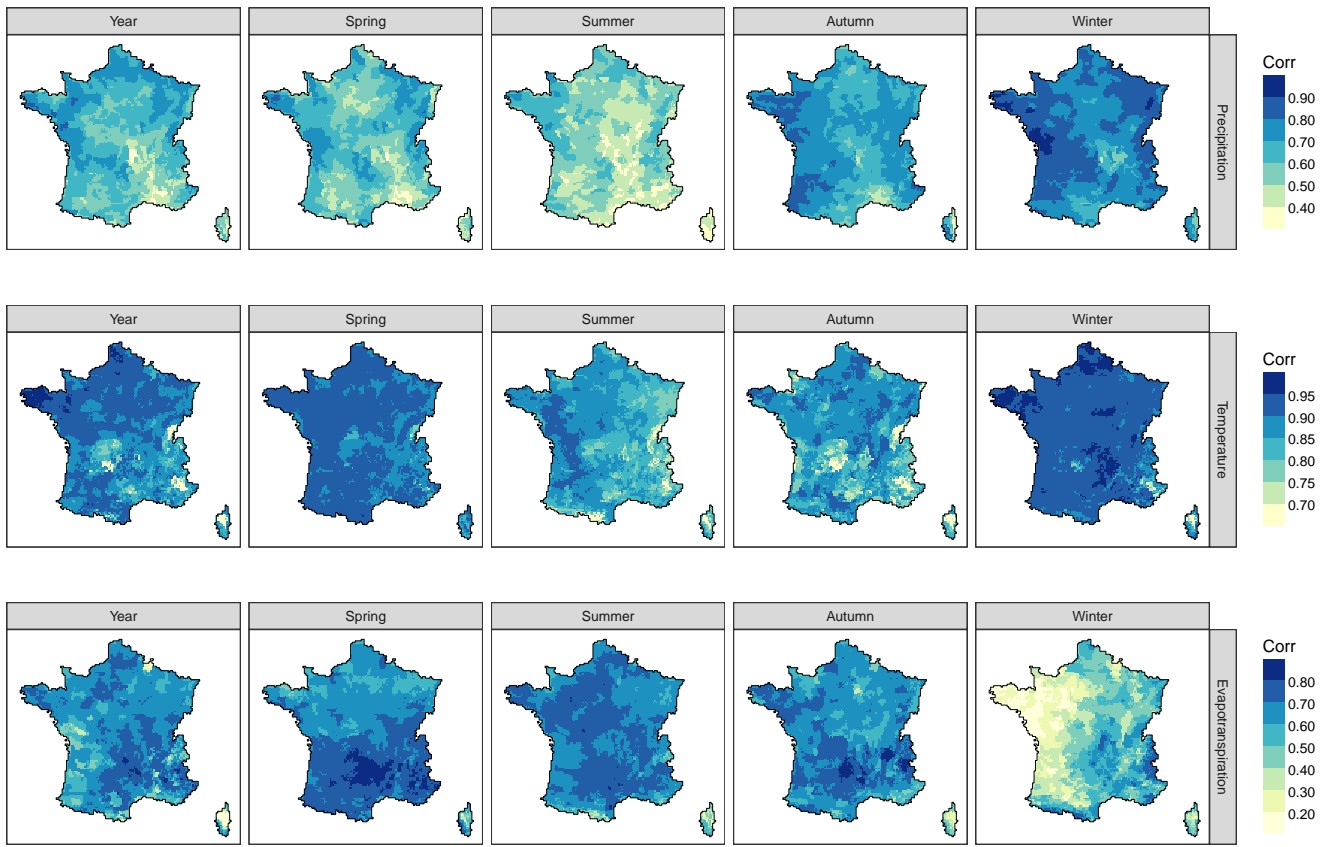

**Figure 10.** Median of annual and seasonal precipitation (top row), temperature (middle row) and reference evapotranspiration (bottom row) correlation between Safran and SCOPE Climate for the 1959–2007 period.

Figure 9 shows the median of annual and seasonal precipitation, temperature and reference evapotranspiration bias between Safran and SCOPE Climate over the 1959-2007 period. The median of annual precipitation bias between Safran and SCOPE Climate shows an absolute value under 5% for the entire France, except for a specific area in the Cévennes region (top row of Fig. 9). In summer, autumn and winter, the bias is generally under 5% in absolute value, and up to ±15% for specific

10    areas. Spring precipitation is overestimated for the Atlantic coast and the Mediterranean region. As spring is the only season when there are a few dry analogue dates after SANDHY-SUB (see Sect. 2.2.2), this could be resolved by adapting the step



of the SCOPE method for each season. Nevertheless, for the sake of parsimony and because the bias correction is already adapted to each zone, the choice was made to have the same parameters for all seasons. The median of annual and seasonal temperature bias is low (middle row of Fig. 9). It is limited to $\pm 0.20°$C at the annual scale and generally stays under $\pm 0.60°$C at the seasonal scale. The maximum bias is reached for a small region in the north-west of France and around Paris with an

overestimation of $+0.63°$C in winter. The median evapotranspiration bias is kept under $\pm 10\%$ at the annual time scale, and largely under $\pm 20\%$ in spring and summer. This bias is higher in autumn and especially in winter with an over-estimation of evapotranspiration between 30% and 40% for the north-east of France (up to 50% for Brittany). The SCOPE method has not been adapted for reference evapotranspiration specifically, and the results obtained for this variable should therefore be taken with care. Evapotranspiration biases are due to a too large selection of analogue days in other seasons than the target day

season. This feature, strongly present in SANDHY outputs, has been reduced with the addition of the stepwise subselection (see Sect. 2.2.2). Nevertheless, this is not enough for removing the biases in reference evapotranspiration.

The median of annual correlations between Safran precipitation and SCOPE Climate precipitation over the 1959-2007 period shows values above 0.6 except for the south-east of France (Fig. 10). Correlations are similar for the spring season, but are lower for summer. This can be explained by the numerous local storms occurring in summer, and which are hardly predictable

by the SCOPE method. Precipitation correlations are generally above 0.8 in autumn and winter. The median of correlations for temperature shows values generally above 0.8-0.9 for the annual and seasonal time steps, with only a few zones in autumn showing lower values. For reference evapotranspiration, the median correlations are generally above 0.6 for all time steps, except for winter when a large part of France shows low correlations. This last result is consistent with results from Fig. 8.

The Continuous Ranked Probability Score (CRPS, Brown, 1974; Matheson and Winkler, 1976) is used to compute a

probabilistic evaluation of SCOPE Climate at a daily, monthly, and annual time step. This score is equivalent to the mean absolute error in a deterministic context, and is normalized by a climatological reference score to allow for comparing zones with different climates. The resulting Continuous Ranked Probability Skill Score (CRPSS, see Sect. 3.4 in Caillouet et al., 2016, for more details) is shown in Fig. 11 for all variables of SCOPE Climate and all time steps. Positive CRPSS values denote an improved skill with respect to the climatology, and 1 reflects a perfect reconstruction. For precipitation, the spatial distribution

of the CRPSS is similar for the three time steps. The lowest performances are found for the Mediterranean coast, the east of France, Corsica, and the eastern part of Massif Central. The highest CRPSS values are found for the Atlantic coast, the French Alps, the Cévennes area, and the western parts of the Massif Central, Jura and Vosges mountain ranges. For temperature, the highest scores for the daily and monthly time steps are obtained for mountainous areas while the lowest ones are obtained for the north-west (only for the monthly time step), south-east and Corsica. A much smoother spatial distribution is found at the

daily and monthly time steps compared to the annual time step. At the annual time step, the highest scores are obtained for the north-west of France, and some specific cells in the south-east have a negative CRPSS. For reference evapotranspiration, the spatial distribution of CRPSS is similar at the daily and monthly time steps: The Highest scores are found for the south-east of France, and particularly for the French Alps and the western part of Massif Central. The lowest scores are found for the north-west of France, and the major part of France shows negative values at the monthly time step. This negative skill is partly

explained by the shifted seasonal cycle (see Fig. 7), in addition to the seasonal bias (see Fig. 9). As for temperature, the spatial



**Figure 11.** Daily, monthly and annual CRPSS between SCOPE Climate and Safran for precipitation, temperature and reference evapotranspiration over the 1959-2007 period. Note the different scales across the maps.

distribution of CRPSS at the annual time step shows relatively high spatial discontinuities. The major part of France show positive annual values, except for Corsica where strong negative values are found.

### 3.3 Performance against homogenized times series

The evolution of the spatially-averaged Root Mean Square Error (RMSE) for both Safran and SCOPE Climate, using the
5  homogenized series as a reference, is presented in Fig. 12. Monthly RMSEs are first calculated using the 323 precipitation time series, and the 65 temperature time series. This leads to 1 RMSE per month for Safran, and 25 RMSEs per month for SCOPE Climate. These RMSEs are then averaged yearly for each SCOPE Climate member. The RMSE in mm/year is then





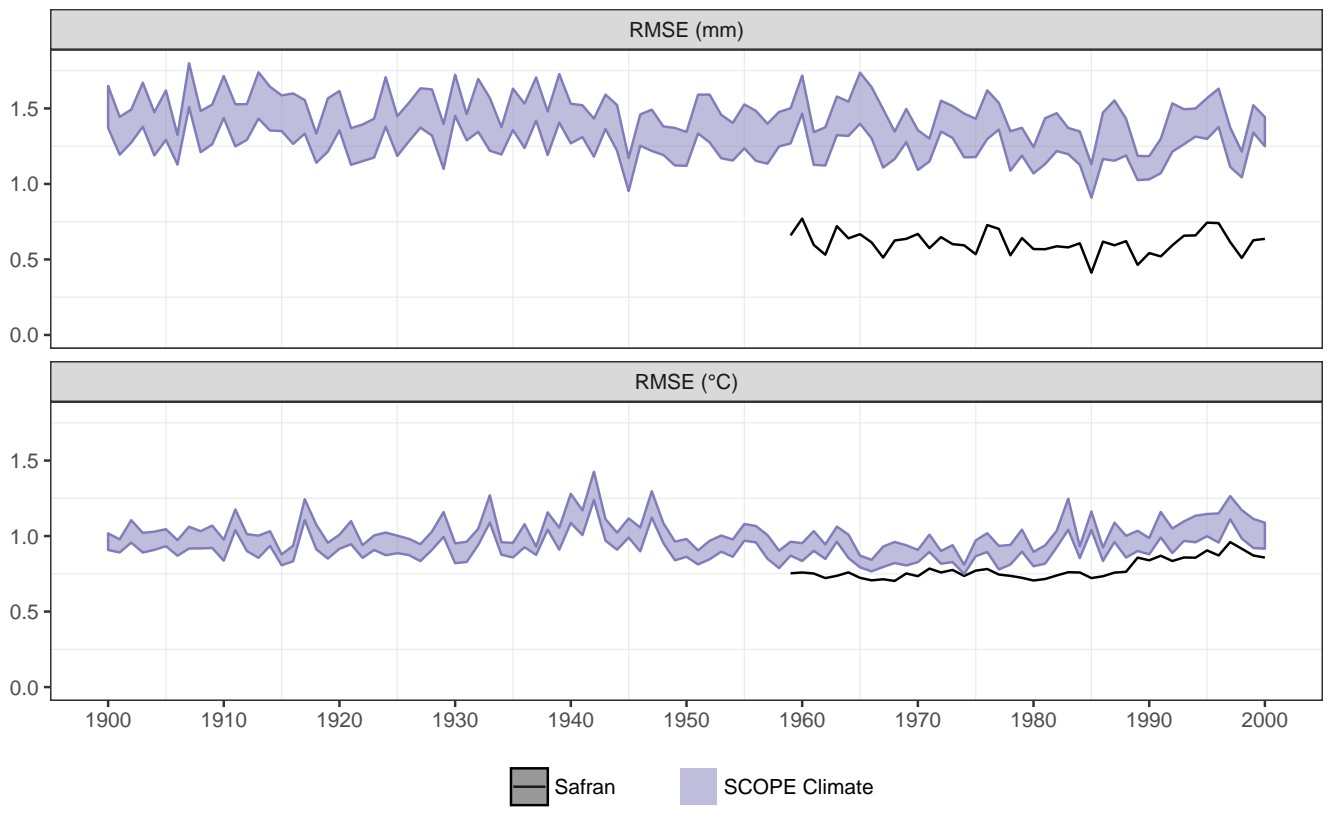

**Figure 12.** Temporal evolution of the precipitation and temperature RMSE for both Safran and SCOPE Climate, with the homogenized series as a reference. Values are initially computed at the monthly time scale. See text for details.

divided by the number of days in each year to get an RMSE in mm/day. Ribbons then show the range between minimum and maximum annual values of the 25 members. For precipitation, RMSEs are relatively constant over time, with an average value around 1.3 mm/day. They are however slightly higher before the 1940s. It is important to note that Safran errors, which have been documented by Vidal et al. (2010), account for nearly half (around 0.6 mm/day) of the reconstruction errors. For

5    temperature, SCOPE Climate RMSEs are around 1°C over the entire period. It even reaches down to 0.9°C over 1959–2000 when Safran errors are around 0.8°C.

## 4    Data availability

SCOPE Climate is 142-year high-resolution ensemble gridded daily meteorological reconstruction of precipitation, temperature and reference evapotranspiration over France. This dataset is the result of the statistical downscaling of 20CR by the SCOPE

10   method. Results are available for 25 members at a daily time step and over a 8-km grid. This dataset is spatially coherent



over France, and has a high inter-variable coherence, thus enabling studies requiring spatial and multi-variable meteorological variables, such as hydrological studies. SCOPE Climate has been used as forcings to create the SCOPE Hydro dataset that provides ensemble daily streamflow time series for for more than 600 near-natural catchments in France over the 1871-2012 period (Caillouet et al., 2017). It has also been used to study spatio-temporal extreme low-flow events in France since 1871 by

Caillouet et al. (2017).

The SCOPE Climate dataset is available under the Attribution-NonCommercial 4.0 International (CC BY-NC 4.0): One may copy and redistribute the material in any medium or format, remix, transform, and build upon the material, until the following terms: *Attribution* – One must give appropriate credit, provide a link to the license, and indicate if changes were made. One may do so in any reasonable manner, but not in any way that suggests the licensor endorses you or your use; *NonCommercial*

– One may not use the material for commercial purposes.

SCOPE Climate is available through the Zenodo repository (http://zenodo.org), as three different datasets, one for each variable (precipitation, temperature, and reference evapotranspiration). Each dataset stores 25 NetCDF files corresponding to the 25 SCOPE Climate members. Please note importantly that when using multiple variables simultaneously, e.g. precipitation and temperature, one may use precipitation from member 1 with temperature from member 1 (and member 2 with member 2),

as SCOPE Climate provides consistent gridded values across variables for any single member. SCOPE Climate can be downloaded as NetCDF files from http://doi.org/10.5281/zenodo.1299760 for precipitation, http://doi.org/10.5281/zenodo.1299712 for temperature, and http://doi.org/10.5281/zenodo.1251843 for reference evapotranspiration.

## 5   Data limitations

The use of SCOPE Climate is conditional on the large-scale information provided by the 20CR atmospheric reanalysis, and

using an alternative extended reanalysis like ERA-20C may lead to different outputs (see Dayon et al., 2015). SCOPE Climate takes into account the uncertainty related to the SCOPE method, but not the uncertainty related to 20CR, as only the ensemble mean has been considered. As discussed by Caillouet et al. (2016), the different members of 20CR show an increased spread before the 1930s/1940s. SCOPE Climate might thus show a lower quality before this period. Similarly, using another statistical downscaling method from either 20CR or ERA20C may lead to different outputs.

Results for reference evapotranspiration showed weak performances at a daily and monthly time step. Thus, it is not recommended to use SCOPE Climate for specific studies on evapotranspiration. Nevertheless, it is possible to use this variable in hydrological modelling for regions and/or temporal periods where/when this variable is not the main driver of streamflow. Caillouet et al. (2017) showed that streamflow reconstructions driven by SCOPE Climate have a high skill for most of the 662 near-natural catchments considered across France.

The adaptation of the Schaake Shuffle procedure in the step 4 of the SCOPE method allowed to retrieve an observed spatial coherence in SCOPE Climate reconstructions. In a forecasting context, the original Schaake Shuffle procedure also allows to retrieve an observed temporal coherence in time series. This feature has not been directly transferred to this reconstruction context as the same number of years should be available in the archive and target periods, which is not possible in this case by



definition. Thus, time series have been constructed by independently combining analogue dates from one day to another and the only temporal coherence is insured by that of the large-scale predictors.

Finally, as the downscaling method is based on the analogue principle, it is not possible to reconstruct higher (for precipitation, temperature and reference evapotranspiration) or lower (for temperature) values than the ones available in the archive
dataset. Nevertheless, values averaged over several days or over several climatically homogeneous zones may exceed these limits. The performance of SCOPE Climate in reconstructing convective precipitation events has not been assessed, and may be lower than for stratiform events as relevant information may not be included in the large-scale predictors of the SCOPE method.

*Author contributions.* L. Caillouet developed the dataset and prepared the manuscript. J.-P. Vidal, E. Sauquet, and B. Graff contributed to
the development of the dataset. J.-M. Soubeyroux supplied the Safran reanalysis. All co-authors contributed to the manuscript.

*Competing interests.* The authors declare that they have no conflict of interest.

*Acknowledgements.* Support for the Twentieth Century Reanalysis Project dataset is provided by the U.S. Department of Energy, Office of Science Innovative and Novel Computational Impact on Theory and Experiment (DOE INCITE) program, and Office of Biological and Environmental Research (BER), and by the National Oceanic and Atmospheric Administration Climate Program Office. The authors would
like to thank Météo-France for providing access to the Safran database. Analyses were performed in R (R Core Team, 2016), with packages dplyr (Wickham and Francois, 2015), ggplot2 (Wickham, 2009), RColorBrewer (Neuwirth, 2014), reshape2 (Wickham, 2007), sp (Pebesma and Bivand, 2005; Bivand et al., 2013), grid (R Core Team, 2016), gridExtra (Auguie, 2016) and scales (Wickham, 2016). Laurie Caillouet's PhD thesis was funded by Irstea and CNR.



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
