# Peer review of "SCOPE Climate: a 142-year daily high-resolution ensemble meteorological reconstruction dataset over France"

_Earth System Science Data, 2018_

## Referee Comment (RC1) · Anonymous Referee #1 · 19 Oct 2018

The article presents a dataset of reconstructed daily meteorological data. Gridded precipitation, temperature, and reference evapotranspiration were reconstructed over 142 years by means of an analogue approach named SCOPE. The article presents the construction of the dataset as well as different welcomed quality evaluations. The dataset can find various useful applications. The figures are of very good quality and the results are interesting. This is an original work worth being published after consideration of some corrections.

- It was highlighted in different studies that 20CR has larger errors that other reanalyses, which is expected due to the small amount of data assimilated. This

will have an impact on the results of the analogue method. Of course, to cover this period, you have no choice but to use 20CR. It is stated that using another reanalysis would results in different predictions, but you should discuss the consequences of using 20CR on the quality of the prediction. This dataset aims at reconstructing accurate past meteorological conditions, not only statistically correct, but also with the correct chronology (am I correct?). Has it been compared with a reconstruction using ERA20C on a shorter period?

- Safran (& 20CR): comment on the quality of the products. What are the known errors and uncertainties? What can be their impact on the final product?

- Different periods: we get a bit confused with the time periods (archive, target, calibration). Can it be summarized clearly? Is there a period for independent validation?

- It is not clear if the domains for the geopotential heights were the ones from Radanovics (as stated in P4 L22-23) or if they were optimized again (as stated in P4 L29).

- Sect. 2.2.2: Please restructure the section. It starts with "The stepwise subselection ..." as we are supposed to know about it, but it is explained in the next paragraph. The definition should come earlier.

- Sect. 2.2.3: When you remove a precipitation analogue and duplicate another one, do the same happen with the temperature and the ET for the same dates? If not, how do you keep the physical consistency between variables? Please specify.

- Sect. 2.2.4: Please explain if the reordering is the same for all variables (P, T, ET) when the ensemble members are reordered, so that their ranks stay consistent. If so, is the order based on the precipitation and applied to the rest? If not, as

previously, how do you keep the physical consistency between variables? Please specify.

- Sect. 3.1.1: Do the analogues to 1910 correspond to other dates with flood events? It would be interesting to know.

- Sect. 3.1.2: Comparison to the station precip: It should be mentioned at the beginning of the work if Safran's gridded precipitation is point precipitation or areal mean precipitation. In case of areal mean precip, comment on the fact of comparing areal mean and station precip.

- Sect. 3.1.2: On the plots of Fig 5, the precip seems to be under-dispersive in summer. It would be desirable, when concluding that the observation falls well into the range of SCOPE climate, to support it with rank histograms.

- Sect. 3.2.1: How did you select the four different cell? Is this setting representative of the rest of the dataset?

- Sect. 3.2.1: Your text sounds like the precipitation in Fig. 8 is good, when you are actually missing most of the main events and are producing peaks when no precip was observed. You state that the sequence of dry and wet periods is well represented, and the bias was fixed. However, if the actual chronology is not accurate, can users really use the dataset to analyse past events, or should it rather be used as a climate simulation (not real chronology)? Is Safran a reliable reference here? Please better discuss the results.

- You try by different ways to reduce the selection of analogues from other seasons. This exchange of seasons causes problems with ET (P18 L7-11). What about coming back to a fixed calendar preselection (moving temporal window)? Does the preselection on temperature really justify adding such complexity to the method (SST and T2m) and having issues with ET? I do not expect a full analysis on this, but it should be discussed.

Technical corrections:

- P1 L15-17: Long sentence. Please rephrase.

- P2 L1-4: Please rephrase.

- P2 L8: Be more specific

- P2 L13: Mentioning "one" of the resulting dataset let us wondering what the others are. Are they equivalent climate reconstructions? If so, why is this one better?

- P2 L15-16: "appropriate space and time resolutions for hydrological applications": it depends on the catchment size and the goal of the application! As any other dataset, it cannot fit all purposes (e.g. flash floods). Please be more specific on which applications are possible.

- P2 L18: "the most general choices": what kind of choices? Be more specific.

- P2 L18-21: This paragraph sounds more like a conclusion than an introduction.

- P3 L16: "...spatially interpolated on the 2.5 deg. grid required..." How did you do the interpolation? Why is it required?

- P4 L7-9: Not clear. Please rephrase.

- P4 L12: Please rephrase.

- P4 L13: "four analogy levels": Better, explain that these are consecutive subsampling steps.

- P4 L14: 4-day window: is 2 days sufficient for the independence of the geopotential height?

- P5 L2: "independently for the 608 climatically homogeneous zones": What do you mean? Please be more specific.

- P5 L6: Improve in what aspect?

- P6 L11: It is not clear when stating "the lowest precipitation" if zeros are included or not.

- P6 L12: "resampled": are they duplicated?

- P6 L13: How is the value of N chosen? Why is three the maximum?

- P6 L27: Why "Julian day"?

- P7 L12: Specify which region

- P10 L7: "heavy amounts": Please rephrase.

- P16 L3-4: Not clear. Please rephrase.

- P17 L11 – P18 L1: Please explain

- P18 L21: The CRPSS is normalized by the climatology, not the CRPS.

- P18 L22: Which climatological reference did you use?

- P18 L31: What is responsible for the irregular patterns and the negative CRPSS at the annual time step?

- P19 L6-7: Please rephrase.

- P21 L30-33: Not clear. Please rephrase.

---

## Referee Comment (RC2) · Anonymous Referee #2 · 29 Oct 2018

Review ESSD-2018-79, reconstructed met data for France

Authors and others have published prior applications, authors themselves have published a separate prior description of SCOPE. Now they provide a careful well-written well-justified description of the SCOPE data product. Sets a good example: country-specific re-analysis followed by careful downscaling of a global reanalysis to reconstruct a long high-resolution meteorological / hydrological record. Useful and necessary expansion of Appendix B in 2017 HESS paper (as the authors explicitly state in line 29 on page 3 of this manuscript). Propose as a 'remedy' to sparse - in duration and location - data, but of course the reanalyses themselves, e.g. 20CR or ERA-20C, depended originally on the same sparse (in space and time) observational networks. Description of the downscaling probably belongs in a different journal but in this case the authors took the initiative and opportunity to validate against, e.g., long local time series.

Two potential uses: for research on hydrology in/over France and, somewhat neglected, as an example for other similar applications in other countries. How would or will SCOPE work in data sparse areas, e.g. Canada or Russia with interesting and vital hydrology in the frozen north but with data and research focus on small agricultural areas of the south? Or Brazil or China for similar reasons - rare reliable data time series from mostly-urban locations needing temporal and spatial extrapolation to larger areas covering a range of land surface types? (Additional similar comment below.)

Overall a good description of an impressive effort. Recommend publication subject to some modest changes / improvements.

Specific comments.

Page 2 line 18: Here the authors use the word 'general' - "The most general choices have been made …" 'General' in this context can unfortunately imply casual, or avoiding specific information. I think the authors mean 'broadly relevant', e.g. that they carefully and intentionally constructed this product to serve a wide range of research users. Replace the word 'general' to better describe their intent?

Page 2 line 19: "the ensemble aspect". The authors clearly intend this as an advantage of this product but for an observational audience in ESSD, they may need to elaborate about those supposed advantages. Later they compare random-selected ensemble members and in one case focus on one specific ensemble member. They even show statistical uncertainties clearly derived across 25 ensemble members, so they clearly regard the ensemble aspect as an asset. Authors need to share their confidence a bit more explicitly here?

Page 2 line 21: "… provides homogeneous time series that will ensure the spatial consistency required for all studies." These authors may themselves 'require' spatial consistency for their own work and may hope that other researchers follow this example. However, one can imagine a range of applications and publications based on this product that will not need or acknowledge consistency with other uses imposed by shared use of one product. Rather than 'required', I think the authors mean 'encouraged' or 'enabled'? Also, spatial consistency in this case derives from 'spatial' homogeneity but as written the sentence allows confusion between spatial and temporal homogeneity and consistency, e.g. a homogeneous time series ensures spatial consistency?

608 climate zones but across the area of France that represents an average climate zone extent of roughly every 30x30km? Thus, at 8 km resolution, perhaps 12-16 grid points in an average climate zone but more likely only one or two grid points in small alpine zones with perhaps 50 grid points per broad climate zones of the Atlantic coast, Mediterranean coast or central agricultural areas (e.g corresponding to the HER regions Armoricain, Mediterraneen or Tables calcaires in Appendix C of the HESS paper)? But only 22 HER (Hydro-ecoregions) used in the HESS paper? Where did the 8 km resolution come from: computational limitations, geospatial considerations? Why focus on 608 regions here but only 22 in the hydrology application? What advantages?

Page 2 line 28 and following: Confusion about Safran temporal extent. Here we read " August 1958 onwards" which implies up to present day. At the end of the paragraph, however, we read " August 1958 to July 2008". But, if Safran relies on "first guess from the ERA-40 reanalysis",

ERA-40 covers only to 2002? Later, on page 4 line 29 we read "using the August 1982-July 2002 period". But, two lines later we again read August 1958 to July 2008. Later still, including in Figure legends, a reader sees 1958 to 2008, 1958 to 2007, 1958 to 2002, etc. At one point later the authors present a set of 2011 data, described as outside of Safran but valid for comparison with Safran? Please can the authors specify the precise data sources and assimilation processes of Safran and consequently its exact temporal extent?

Page 3 line 6 and following: here a reader learns that the time extent of SCOPE comes from the time extent of V2 of 20CR, e.g. 1871 to 2012, because (as stated on Page 4 line 8) "the large-scale reanalysis is then the only dataset that must be available over the period to reconstruct." This information should have come earlier, to justify the SCOPE time period? Authors correct about 20CR depending on SLP but 20 CR V2 also assimilated SST and perhaps sea ice?

Page 3 line 21 and following: the authors used only a single Atlantic SST? Given the maps and the inclusion of Corsica, why did they not also include a Mediterranean SST data point? In the phrase "optimised grid cell", what does 'optimised' mean? I understand why they include an ocean predictor but I do not understand why only this one? Perhaps the only one with a long-enough time series record, and even then they had to interpolate from monthly to daily? So far as I can tell, Appendix B of the HESS paper (cited in line 23) makes no reference to ocean data, no reference to monthly to daily interpolation, and no mention of SST as a valid predictor? Later (line 1 of Page 6) the authors justify this single SST point based on "consistency over France and parsimony of parameters" but additional SST points would not change the number of large-scale predictors and subsequent importance and uncertainty assigned to Mediterranean regions and convective precipitation suggests perhaps the benefit of including Mediterranean ocean SST. No longer possible for this data product, but something to recommend for next version or future efforts?

Page 5 lines 4,5: SANDHY set up for a single predictand, precip. Authors want to extend SANDHY to two additional predictands, temperature and evapotranspiration. Therefore, authors developed the following steps, e.g. 2.2.2, 2.2.3 and 2.2.4? Step 2 (SANDHY-SUB) adds SST and T2M as predictors? Step 3 corrects a 10% dry precip bias by iteratively removing driest analog years (?) followed by resampling? Step 4 involves a rank correlation shuffle within ensemble members to improve spatial cross-correlation but because of shorter archive time period compared to longer target period, the shuffling can only improve spatial correlations, not temporal correlations? That fact that I needed to write this summary for myself suggests that the authors, in their accurate and detailed sequential description, have left this reader a bit in the dark about how each step leads to the overall goal. Perhaps in the final paragraph of the Introduction or as the initial outline of Section 2?

Page 6 Figure 2: is identical to Figure B1 from the HESS paper and should be cited as such?

Page 16 Figure 9 (and applies to other figures): Authors mix percentage differences for precip and evapotranspiration with absolute differences for temperature. But, guessing at an average temperature for France of 10C, 0.2C difference would represent 2%, much better than but now at least in the same units as precip and evapotranspiration? The authors have some reason for using this mixture of relative and absolute units?

Page 17 line 4,5: Valid caution here about application of SCOPE evapotranspiration to specific events but this caution should appear or should also appear in the data limitations section, e.g. page 21 starting from line 25? That paragraph seems to hint at this issue, but this statement provides a more specific example.

page 17 line 7,8: Here the authors provide precip bias estimate "median of annual precipitation bias between Safran and SCOPE Climate shows an absolute value under 5% for the entire France , …". But on Page 6 line 16 the authors claimed "retrieve a near-zero bias in mean interannual precipitation over France." Respectable results in either case, but do these two statements coincide or differ?

Page 17 line 10,11: Confusing! Does this statement "… spring is the only season when there are a few dry analogue dates after SANDHY-SUB …" indicate that the subsetting removed dry days

from the spring, leaving fewer dry days and a larger proportion of normal or wet days?  Or, does the statement indicate - as written but seems unlikely - that only spring has a few dry days?  Please revise and clarify.

Page 22 line 6: convective precipitation.  For this reader, the authors have offered several hints throughout the manuscript of this combined temporal, resolution and geographic weakness for convective precipitation, vis. weaknesses in Mediterranean regions, weaknesses in abrupt topographic regions, weaknesses in Atlantic SST as applied to Mediterranean coastline, weaknesses in autumnal precip and temperature patterns.  These various factors, if highlighted and combined here, would add some specificity to the convective precip issue?  Rather than weakening the outcome, identification of this process as difficult would in fact strengthen the reader's sense that the authors know their product and their French geography.  A sentence about this difficult-to-resolve issue should appear in the abstract?

Somewhere, probably in the conclusions, the authors could / should include a sentence or two about application of the SCOPE approach, and specifically of the three 'improvements' imposed after the SANDHY step, to other settings.  The authors have noted the positive aspects of high-resolution time series available for the UK.  Could they here make a comment about the opposite situation: larger areas with scarce or no data?  Could they or anyone who wishes to reproduce their effort for other regions even imagine this work without the existence of Safran?  How would a country of large area and broad range of land surfaces develop information and skills necessary to repeat a SCOPE-like reconstruction?  Works nicely for France.  Could it work elsewhere?

---

## Author Comment (AC1) · 23 Nov 2018

*The article presents a dataset of reconstructed daily meteorological data. Gridded precipitation, temperature, and reference evapotranspiration were reconstructed over 142 years by means of an analogue approach named SCOPE. The article presents the construction of the dataset as well as different welcomed quality evaluations. The dataset can find various useful applications. The figures are of very good quality and the results are interesting. This is an original work worth being published after consideration of some corrections.*

The authors would like to thank Referee 1 for his/her strong positive comments on the manuscript. We also thank him/her for the specific and technical comments (in italic below) that will lead to improve the manuscript. The detailed answers to the specific comments are presented below.

*• It was highlighted in different studies that 20CR has larger errors that other reanalyses, which is expected due to the small amount of data assimilated. This will have an impact on the results of the analogue method. Of course, to cover this period, you have no choice but to use 20CR. It is stated that using another reanalysis would results in different predictions, but you should discuss the consequences of using 20CR on the quality of the prediction. This dataset aims at reconstructing accurate past meteorological conditions, not only statistically correct, but also with the correct chronology (am I correct?). Has it been compared with a reconstruction using ERA20C on a shorter period?*

Correct chronology is indeed aimed at in SCOPE Climate. Literature findings about inhomogeneities of 20CR predictors and associated uncertainties that may affect SCOPE Climate are provided in Sect. 6.1 of Caillouet et al. (2016). Inhomogeneities have been found in the ensemble mean of 20CR before the 1930s, because of the too few data assimilated in this period. Consequently, SCOPE Climate may be less accurate before the 1930s. Some sentences will be added in the discussion section to specify this. ERA-20C has been released at the end of our work, so no comparison between these two global reanalyses has been done. Nevertheless, Dayon (2015) provided a quick comparison of ERA-20C and 20CR for downscaling purposes, only mentioning lower scores for spring temperatures reconstructed with 20CR in comparison to ERA-20C. A more recent study focused on the impact of the driving reanalysis on downscaled precipitation over Switzerland with various analogue methods, including SANDHY (Horton et al., 2018)

*• Safran (& 20CR): comment on the quality of the products. What are the known errors and uncertainties? What can be their impact on the final product?*

See the above response for 20CR. Vidal et al. (2010) performed a detailed validation of the gridded Safran dataset with both dependent and independent data. They showed that the errors on precipitation are low and constant over the 1958–2008 period. Errors on temperature are decreasing with the increasing number of available surface observations. This will be added to the main text. We therefore believe that possible temporal inhomogeneities of SCOPE Climate mainly come from 20CR, as possible inhomogeneities in Safran are smoothed out through the multi-member analogue resampling.

*• Different periods: we get a bit confused with the time periods (archive, target, calibration). Can it be summarized clearly? Is there a period for independent validation?*

This will be rephrased. As there is no calibration period, there is no validation period. SANDHY needs two periods: a target period, which is the period to reconstruct, and an archive period, which the period when the meteorological situations are picked up to reconstruct the target period. Large scale reanalysis (20CR) should be available on both periods whereas local scale reanalysis (Safran) should be available on the archive period. For a date in the target period, large-scale situation will be analysed. The most similar situations will be found in the archive period and correspondent local

meteorology will be used to reconstruct the meteorology of the picked up date in the target period. To produce SCOPE Climate, target period is 01/01/1871-31/12/2012 and archive period is 01/08/1958-31/07/2008. Moreover, in SANDHY, geopotential analogy domains are optimised with an algorithm of growing rectangular domains over the 1982-2002 period. SCOPE Climate has been validated using homogenized time series on the 1900-2000 period.

*• Sect. 2.2.2: Please restructure the section. It starts with "The stepwise subselection . . ." as we are supposed to know about it, but it is explained in the next paragraph. The definition should come earlier.*

Indeed, this will be restructured.

*• Sect. 2.2.3: When you remove a precipitation analogue and duplicate another one, do the same happen with the temperature and the ET for the same dates? If not, how do you keep the physical consistency between variables? Please specify.*

Indeed, as analogue days are the same for precipitation and temperature, if the day is removed after correction of the precipitation bias, it will also be removed for temperature. Physical consistency is therefore kept. This will be added to the text.

*• Sect. 2.2.4: Please explain if the reordering is the same for all variables (P, T, ET) when the ensemble members are reordered, so that their ranks stay consistent. If so, is the order based on the precipitation and applied to the rest? If not, as previously, how do you keep the physical consistency between variables? Please specify.*

For the Schaake Shuffle part, reordering is done independently for each variable. Nevertheless, observed rank correlations are derived from the Safran multivariate meteorological fields and applied to the reconstructed ensemble. It thus ensures a spatial and intervariable coherence of any single ensemble member. This physical consistency is therefore at least as good as in Safran.

 *• Sect. 3.1.1: Do the analogues to 1910 correspond to other dates with flood events? It would be interesting to know.*

This would indeed be very interesting to look out. This is presumably the case, as it is the basis for analogue downscaling, provided that analogue dates are sufficiently grouped in time to generate a flood on the very slow-reactive Seine basin. Such a verification is of course outside of the scope of this study.

*• Sect. 3.1.2: Comparison to the station precip: It should be mentioned at the beginning of the work if Safran's gridded precipitation is point precipitation or areal mean precipitation. In case of areal mean precip, comment on the fact of comparing areal mean and station precip.*

Indeed, it is mentioned p.2 l.29 that Safran is a gridded product at an 8-km resolution, representing mean precipitation over the grid cell. Reality is actually a bit more complicated than that: gridded precipitation (and other variables) in Safran are values at the altitude of the grid cell, and an average over the climatically homogenous zone containing the grid cells (see Vidal et al., 2010 for details) is done. Comparison between gridded products like Safran or SCOPE Climate and station data are of course not fair.  This fact will be added to the text.

*• Sect. 3.1.2: On the plots of Fig 5, the precip seems to be under-dispersive in summer. It would be desirable, when concluding that the observation falls well into the range of SCOPE climate, to support it with rank histograms.*

Summer precipitation from SCOPE might be under-dispersive, but rank histograms are not necessarily well suited in this particular case, given the low sample size of years. Below is plotted the rank histogram for summer precipitation with respect to the homogenized series. It is very noisy and interpreting it is not straightforward. We would therefore prefer not to include such rank histograms in the manuscript.

[Figure]

• *Sect. 3.2.1: How did you select the four different cell? Is this setting representative of the rest of the dataset?*

The four different cells have been selected in four zones with different climatic influences. Oceanic for Finistère, mountainous for Haute Savoie, meridional oceanic for Corrèze and Mediterranean for Cévennes. Regimes from different cells in these zones have been analysed and the ones being the most different from each other have been kept. Not all the cells in France have been studied but we believe that these 4 cells represent a good subset of the rest of the dataset.

• *Sect. 3.2.1: Your text sounds like the precipitation in Fig. 8 is good, when you are actually missing most of the main events and are producing peaks when no precip was observed. You state that the sequence of dry and wet periods is well represented, and the bias was fixed. However, if the actual chronology is not accurate, can users really use the dataset to analyse past events, or should it rather be used as a climate simulation (not real chronology)? Is Safran a reliable reference here? Please better discuss the results.*

Well, the figure in its present state may be misleading in that respect, and we'll be working on adapting it to better convey the actual performance of SCOPE Climate. The actual figures on wet/dry days with a threshold of 0.1 mm are the following, for the year 2011 and this cell: 131 potentially wet days in SCOPE (with at least one member being wet) out of 141 (i.e. 92 %), and 221 potentially dry days in SCOPE (with at least one member being dry) out of 223 (i.e. 99%). So one may say that the chronology is rather accurate. On the question of Safran being a reliable reference: Safran is the gridded reference product over France, with its advantages and limitations of being a gridded

product. In this specific comparison, the comparison is rather fair, as we compare two gridded products, SCOPE being a resample of Safran data from the 1958-2008 period.

• *You try by different ways to reduce the selection of analogues from other seasons. This exchange of seasons causes problems with ET (P18 L7-11). What about coming back to a fixed calendar preselection (moving temporal window)? Does the preselection on temperature really justify adding such complexity to the method (SST and T2m) and having issues with ET? I do not expect a full analysis on this, but it should be discussed.*

Indeed, coming back to a seasonal calendar would fix issues about evapotranspiration. Nevertheless, Ben Daoud (2011) showed that allowing a selection of analogues in other seasons for SANDHY is leading to improvements in the downscaling of precipitation. Moreover, setting fixed seasons would not be necessarily ideally suitable for our case – a long-term historical reconstruction – as there could be season shifts. Additionally, tests have been done in a previous paper for having a calendar subsetting in lieu of the stepwise subsetting, and we showed that the stepwise selection leads to higher rank correlations and lower mean errors between reconstructed and observed temperatures (Caillouet et al., 2016).

*Technical corrections*

• *P1 L15-17: Long sentence. Please rephrase.*

This will be rephrased.

• *P2 L1-4: Please rephrase.*

This will be rephrased.

• *P2 L8: Be more specific*

This will be rephrased.

• *P2 L13: Mentioning "one" of the resulting dataset let us wondering what the others are. Are they equivalent climate reconstructions? If so, why is this one better?*

The other resulting dataset is SCOPE Hydro. It has been created using SCOPE Climate in a hydrological model to provide a reconstruction of streamflow for France over the 1871-2012 period. Another paper will make SCOPE Hydro available.

• *P2 L15-16: "appropriate space and time resolutions for hydrological applications": it depends on the catchment size and the goal of the application! As any other dataset, it cannot fit all purposes (e.g. flash floods). Please be more specific on which applications are possible.*

This will be specified.

• *P2 L18: "the most general choices": what kind of choices? Be more specific.*

The methodological choices made to create SCOPE Climate (sub selection, bias correction, Schaake Shuffle) never favored the reconstruction of specific types of events, such as floods or droughts. This will be clarified.

• *P2 L18-21: This paragraph sounds more like a conclusion than an introduction.*

It appeared important for the authors to highlight the innovation introduced by SCOPE Climate in the introduction

• *P3 L16: ". . .spatially interpolated on the 2.5 deg. grid required. . ." How did you do the interpolation? Why is it required?*

It is required as the SANDHY method had been set-up for inputs at 2.5 deg grid. The interpolation is the commonly used bilinear interpolation.

• *P4 L7-9: Not clear. Please rephrase.*

This will be rephrased.

• *P4 L12: Please rephrase.*

This will be rephrased.

• *P4 L13: "four analogy levels": Better, explain that these are consecutive subsampling steps.*

This will be added.

• *P4 L14: 4-day window: is 2 days sufficient for the independence of the geopotential height?*

XXX

• *P5 L2: "independently for the 608 climatically homogeneous zones": What do you mean? Please be more specific.*

This means that the SANDHY method is applied 608 times, each time corresponding to one zone. There is no spatial consistency between zones, the latter being added with the Schaake Shuffle process.

• *P5 L6: Improve in what aspect?*

This is specified l.7, to improve the fact that precipitation is over estimated in spring and under estimated in autumn for Mediterranean areas. As suggested before, this section will be re-ordered.

• *P6 L11: It is not clear when stating "the lowest precipitation" if zeros are included or not.*

They are included. This will be added.

• *P6 L12: "resampled": are they duplicated?*

Indeed, they are duplicated.

• *P6 L13: How is the value of N chosen? Why is three the maximum?*

Median rank of Safran precipitation in the range of SCOPE Climate was 0.55 before removing the bias. To get a rank back at 0.5, this meant removing N=1.25 days. Tests have been done by removing N=1, N=2 or N=3 days for the entire France. Results show that removing these numbers of days -- depending on the zone in France -- allowed retrieving an annual precipitation bias around 0%. Removing more than 3 days for the zones with N=3 would have transformed the slight positive bias in slight negative bias.

• *P6 L27: Why "Julian day"?*

This is simply the day-of-the-year.

• *P7 L12: Specify which region*

This will be specified.

*• P10 L7: "heavy amounts": Please rephrase.*

This will be rephrased.

*• P16 L3-4: Not clear. Please rephrase.*

This will be rephrased.

*• P17 L11 – P18 L1: Please explain.*

Annual precipitation is under estimated, this is the reason why the bias correction consists in removing dry days in the 25 ensemble member of SCOPE Climate. Summer, autumn and winter are indeed under-estimated. Nevertheless, the spring season shows an overestimation of precipitation. This means than removing dry days in spring will potentially exacerbate the precipitation overestimation during this season. This could be managed by adapting the number of dry days to remove depending on the season, as it is done for the zones. Zero dry days would be removed in spring, whereas 1, 2 or 3 days will be removed in the other seasons. This has not been done for the sake of parsimony as the number of days is already adapted to each of the 608 zones.

*• P18 L21: The CRPSS is normalized by the climatology, not the CRPS.*

This will be corrected.

*• P18 L22: Which climatological reference did you use?*

The climatology is calculated using data from Safran over the archive period (1958-2008). Meteorological ensemble data is selected randomly from ±60 days around the target date to take seasonality into account.

*• P18 L31: What is responsible for the irregular patterns and the negative CRPSS at the annual time step?*

These patterns interestingly come from temporal non-homogeneities in the Safran data. More precisely, specific zones with negative CRPSS show temperature trends that are not consistent with surrounding areas, and even sometimes negative over the last decades. This results from changes in the observation network, with more stations being installed at higher altitudes, hence with a lower mean temperature. As shown by Vidal et al. (2010), Safran is not suited for trend analysis. These irregular patterns thus therefore suggest that interannual variability and trends are more spatially homogenous in SCOPE Climate than in Safran. This is however still to be shown.

*• P19 L6-7: Please rephrase.*

This will be rephrased.

*• P21 L30-33: Not clear. Please rephrase.*

This will be rephrased.

References

Ben Daoud, A., Sauquet, E., Lang, M., Bontron, G., and Obled, C.: Precipitation forecasting through an analog sorting technique: a comparative study, Adv. Geosci., 29, 103–107, doi:10.5194/adgeo-29-103-2011, 2011.

Caillouet, L., Vidal, J.-P., Sauquet, E. and Gradd, B.: Probabilistic precipitation and temperature downscaling of the twentieth century reanalysis over France. Clim. Past, 12, 635–662, doi: 10.5194/cp-12-635-2016, 2016.

Dayon, G.: Evolution du cycle hydrologique sur la France au cours des prochaines décennies. PhD thesis, Université Paul Sabatier, Toulouse, 2015.

Vidal, J.-P., Martin, E., Franchistéguy, L., Baillon, M., and Soubeyroux, J.-M.: A 50-year high-resolution atmospheric reanalysis over France with the Safran system, Int. J. Climatol., 30, 1627– 1644, doi:10.1002/joc.2003, 2010

---

## Author Comment (AC2) · 23 Nov 2018

*Authors and others have published prior applications, authors themselves have published a separate prior description of SCOPE. Now they provide a careful well-written well-justified description of the SCOPE data product. Sets a good example: country-specific re-analysis followed by careful downscaling of a global reanalysis to reconstruct a long high-resolution meteorological / hydrological record. Useful and necessary expansion of Appendix B in 2017 HESS paper (as the authors explicitly state in line 29 on page 3 of this manuscript). Propose as a 'remedy' to sparse - in duration and location - data, but of course the reanalyses themselves, e.g. 20CR or ERA-20C, depended originally on the same sparse (in space and time) observational networks. Description of the downscaling probably belongs in a different journal but in this case the authors took the initiative and opportunity to validate against, e.g., long local time series.*

*Two potential uses: for research on hydrology in/over France and, somewhat neglected, as an example for other similar applications in other countries. How would or will SCOPE work in data sparse areas, e.g. Canada or Russia with interesting and vital hydrology in the frozen north but with data and research focus on small agricultural areas of the south? Or Brazil or China for similar reasons - rare reliable data time series from mostly-urban locations needing temporal and spatial extrapolation to larger areas covering a range of land surface types? (Additional similar comment below.)*

*Overall a good description of an impressive effort. Recommend publication subject to some modest changes / improvements.*

The authors would like to thank Referee 2 for his/her positive comments on the manuscript. Indeed, the method has not been applied in other countries – yet. SCOPE requires a high-resolution reanalysis, which will be extended on the longer period of a large-scale reanalysis. In France, the good-quality Safran reanalysis was available, but it is not the case everywhere. Applying SCOPE would not be possible without a reasonably long dataset (20 to 30 years) providing local meteorology. Moreover, large-scale reanalyses, such as 20CR, are of sufficient good quality in France thanks to the high amount of data assimilated over Western Europe. It is not the case everywhere in the world and this would have direct implications on the quality of the reconstructions. This would still be a very interesting extension to the validation of SCOPE to use it in other countries, with the available observations. We also thank him/her for the specific and technical comments (in italic below) that will lead to improve the manuscript. The detailed answers to the specific comments are presented below.

*Page 2 line 18: Here the authors use the word 'general' - "The most general choices have been made …" 'General' in this context can unfortunately imply casual, or avoiding specific information. I think the authors mean 'broadly relevant', e.g. that they carefully and intentionally constructed this product to serve a wide range of research users. Replace the word 'general' to better describe their intent?*

Indeed, this is a good remark. This will be rephrased.

*Page 2 line 19: "the ensemble aspect". The authors clearly intend this as an advantage of this product but for an observational audience in ESSD, they may need to elaborate about those supposed advantages. Later they compare random-selected ensemble members and in one case focus on one specific ensemble member. They even show statistical uncertainties clearly derived across 25 ensemble members, so they clearly regard the ensemble aspect as an asset. Authors need to share their confidence a bit more explicitly here?*

Indeed, randomly selected members have sometimes been selected in order to provide examples of a full clear validation of SCOPE Climate. As the dataset has numerous dimensions (8602 grid cells, daily for 140 years, 3 variables, 25 members), choices to reduce some dimensions have been made to make the validation hopefully easier to understand. Some sentences will be added to the text to explain why it is important to keep an ensemble aspect.

*Page 2 line 21: "… provides homogeneous time series that will ensure the spatial consistency required for all studies." These authors may themselves 'require' spatial consistency for their own work and may hope that other researchers follow this example. However, one can imagine a range of applications and publications based on this product that will not need or acknowledge consistency with other uses imposed by shared use of one product. Rather than 'required', I think the authors mean 'encouraged' or 'enabled'? Also, spatial consistency in this case derives from 'spatial' homogeneity but as written the sentence allows confusion between spatial and temporal homogeneity and consistency, e.g. a homogeneous time series ensures spatial consistency?*

Indeed, if a study focuses on a specific grid cell, the spatial consistency is not required. Nevertheless, if a study is focused on an area composed of several grid cells spanning different climatically homogenous zones, spatial consistency is often compulsory. Spatial consistency allows to study a specific meteorological event in a coherent way over a region or to use the meteorological fields as input to a hydrological model. The authors will think about a rephrasing.

*608 climate zones but across the area of France that represents an average climate zone extent of roughly every 30x30km? Thus, at 8 km resolution, perhaps 12-16 grid points in an average climate zone but more likely only one or two grid points in small alpine zones with perhaps 50 grid points per broad climate zones of the Atlantic coast, Mediterranean coast or central agricultural areas (e.g corresponding to the HER regions Armoricain, Mediterraneen or Tables calcaires in Appendix C of the HESS paper)? But only 22 HER (Hydro-ecoregions) used in the HESS paper? Where did the 8 km resolution come from: computational limitations, geospatial considerations? Why focus on 608 regions here but only 22 in the hydrology application? What advantages?*

The gridded resolution of 8-km comes from the Safran local reanalysis. There are 8602 Safran cells dispatched into 608 climatically homogenous zones, indeed varying in size according to the spatial heterogeneity of e.g. relief (see Vidal et al., 2010, for a map of climatically homogenous zones). The aim of SCOPE is to extend it to the longer period of a large-scale reanalysis (here 20CR). The resulting dataset (SCOPE Climate) will have the same resolution than the original local-scale reanalysis (so 8-km). The hydro ecoregions were used in the HESS paper only as an intermediate spatial scale to simplify the spatial aggregation of low-flow events across France.

*Page 2 line 28 and following: Confusion about Safran temporal extent. Here we read " August 1958 onwards" which implies up to present day. At the end of the paragraph, however, we read " August 1958 to July 2008". But, if Safran relies on "first guess from the ERA-40 reanalysis", ERA-40 covers only to 2002? Later, on page 4 line 29 we read "using the August 1982-July 2002 period". But, two lines later we again read August 1958 to July 2008. Later still, including in Figure legends, a reader sees 1958 to 2008, 1958 to 2007, 1958 to 2002, etc. At one point later the authors present a set of 2011 data, described as outside of Safran but valid for comparison with Safran? Please can the authors specify the precise data sources and assimilation processes of Safran and consequently its exact temporal extent?*

Safran is updated each year and is available 1958 - onwards. Indeed, ERA-40 is ending in 2002. Since 2002, first guest field are not taken from ERA-40 but from ECMWF operational archives (Vidal et al., 2010). To sum up, Safran is available from 1958-onwards, the archive period where local scenarios

are picked up is 1958-2008, the target period to reconstruct is 1871-2012 and the period where analogy domains are optimised is 1982-2002. As the target period is 1871-2012, SCOPE Climate is available on 1871-2012. This is why the authors are able to show a reconstruction of the 2011 year, which is outside of the optimisation and archive periods of the SCOPE method.

*Page 3 line 6 and following: here a reader learns that the time extent of SCOPE comes from the time extent of V2 of 20CR, e.g. 1871 to 2012, because (as stated on Page 4 line 8) "the largescale reanalysis is then the only dataset that must be available over the period to reconstruct." This information should have come earlier, to justify the SCOPE time period? Authors correct about 20CR depending on SLP but 20 CR V2 also assimilated SST and perhaps sea ice?*

As the method is described after the data section, the authors do not think it would be valuable justifying a part of the method in the data section or the introduction. Nevertheless, the introduction specify the period of availability of SCOPE Climate. Indeed for 20CR, this will be added.

*Page 3 line 21 and following: the authors used only a single Atlantic SST? Given the maps and the inclusion of Corsica, why did they not also include a Mediterranean SST data point? In the phrase "optimised grid cell", what does 'optimised' mean? I understand why they include an ocean predictor but I do not understand why only this one? Perhaps the only one with a longenough time series record, and even then they had to interpolate from monthly to daily? So far as I can tell, Appendix B of the HESS paper (cited in line 23) makes no reference to ocean data, no reference to monthly to daily interpolation, and no mention of SST as a valid predictor? Later (line 1 of Page 6) the authors justify this single SST point based on "consistency over France and parsimony of parameters" but additional SST points would not change the number of large-scale predictors and subsequent importance and uncertainty assigned to Mediterranean regions and convective precipitation suggests perhaps the benefit of including Mediterranean ocean SST. No longer possible for this data product, but something to recommend for next version or future efforts?*

One grid point has been chosen for the sake of simplicity. Three grid points have first been studied (North Sea, Atlantic, Mediterranean Sea). The grid point in Atlantic is the one having the largest influence over the territory, even if it was not the best for Corsica. Optimised means that this grid point has been chosen after an optimisation process looking at the correlation between Safran data and all the available grid points. Appendix B of the HESS paper does not talk about the SST predictor as the addition of this predictor is the subject of the entire first article in Climate of the past (Caillouet et al., 2016), and is integrated in the method called "Stepwise". Data for SST comes from ERSSTv3b (see Sect. 2.1.3), which is a reanalysis available between 1854 and present days. Choosing two different grid points for the SST predictor would mean partitioning France for each grid point. This would have raised the question of continuity between these areas. Even with the use of a single grid point, we were able to correct the regime asymmetry of precipitation in Mediterranean areas (with the grid point in Atlantic) as well as the temperature regime (see Caillouet et al., 2016). Further tests might be performed to see the influence of two grid points instead of one.

*Page 5 lines 4,5: SANDHY set up for a single predictand, precip. Authors want to extend SANDHY to two additional predictands, temperature and evapotranspiration. Therefore, authors developed the following steps, e.g. 2.2.2, 2.2.3 and 2.2.4? Step 2 (SANDHY-SUB) adds SST and T2M as predictors? Step 3 corrects a 10% dry precip bias by iteratively removing driest analog years (?) followed by resampling? Step 4 involves a rank correlation shuffle within ensemble members to improve spatial cross-correlation but because of shorter archive time period compared to longer target period, the shuffling can only improve spatial correlations, not temporal correlations? That fact that I needed to*

*write this summary for myself suggests that the authors, in their accurate and detailed sequential description, have left this reader a bit in the dark about how each step leads to the overall goal. Perhaps in the final paragraph of the Introduction or as the initial outline of Section 2?*

Figure 2 is here to summarise all these steps and put them together. We can add some lines near this figure to precise the succession of steps. Concerning the Schaake Shuffle, the initial method allows to ensure the temporal consistency because the method uses a reference dataset with the same length of the reconstructed dataset. This is not possible for us as our reconstructed dataset is 142 years length and the maximum length of our reference dataset is approx. 50 years. Several attempts have been made to adapt this feature, without success for now.

*Page 6 Figure 2: is identical to Figure B1 from the HESS paper and should be cited as such?*

Indeed, this is just the figure summarising all steps. We do not think adding a reference to the HESS paper would help the reader as this paper is mainly for the drought method and not SCOPE.

*Page 16 Figure 9 (and applies to other figures): Authors mix percentage differences for precip and evapotranspiration with absolute differences for temperature. But, guessing at an average temperature for France of 10C, 0.2C difference would represent 2%, much better than but now at least in the same units as precip and evapotranspiration? The authors have some reason for using this mixture of relative and absolute units?*

It is simply not possible to define a percentage difference on a variable that is not always positive like T. Relative difference is better suited for P as they allow for removing the effect of baseline spatial heterogeneity.

*Page 17 line 4,5: Valid caution here about application of SCOPE evapotranspiration to specific events but this caution should appear or should also appear in the data limitations section, e.g. page 21 starting from line 25? That paragraph seems to hint at this issue, but this statement provides a more specific example.*

It appears in page 21 l.25: "Results for reference evapotranspiration showed weak performances at a daily and monthly time step. Thus, it is not recommended to use SCOPE Climate for specific studies on evapotranspiration. Nevertheless, it is possible to use this variable in hydrological modelling for regions and/or temporal periods where/when this variable is not the main driver of streamflow."

*17 line 7,8: Here the authors provide precip bias estimate "median of annual precipitation bias between Safran and SCOPE Climate shows an absolute value under 5% for the entire France , …". But on Page 6 line 16 the authors claimed "retrieve a near-zero bias in mean interannual precipitation over France." Respectable results in either case, but do these two statements coincide or differ?*

These two statements are indeed for the same dataset. The absolute value is under 5% for the entire France, meaning that values can be positive or negative. It leads, if we do the mean, to a near-zero bias.

*Page 17 line 10,11: Confusing! Does this statement "… spring is the only season when there are a few dry analogue dates after SANDHY-SUB …" indicate that the subsetting removed dry days from the spring, leaving fewer dry days and a larger proportion of normal or wet days? Or, does the statement indicate - as written but seems unlikely - that only spring has a few dry days? Please revise and clarify.*

Annual precipitation is underestimated, this is the reason why the bias correction consists in removing dry days in the 25-member ensemble of SCOPE Climate. Summer, autumn and winter precipitation amounts are indeed underestimated. Nevertheless, the spring season shows an

overestimation of precipitation. This means than removing dry days in spring will potentially exacerbate the precipitation overestimation during this season. This could be managed by adapting the number of dry days to remove depending on the season, as it is done for the zones. Zero dry days would be removed in spring, whereas 1, 2 or 3 days will be removed in the other seasons. This has not been done for the sake of parsimony as the number of days is already adapted to each of the 608 zones. This will be clarified.

*Page 22 line 6: convective precipitation. For this reader, the authors have offered several hints throughout the manuscript of this combined temporal, resolution and geographic weakness for convective precipitation, vis. weaknesses in Mediterranean regions, weaknesses in abrupt topographic regions, weaknesses in Atlantic SST as applied to Mediterranean coastline, weaknesses in autumnal precip and temperature patterns. These various factors, if highlighted and combined here, would add some specificity to the convective precip issue? Rather than weakening the outcome, identification of this process as difficult would in fact strengthen the reader's sense that the authors know their product and their French geography. A sentence about this difficult-to-resolve issue should appear in the abstract?*

More than the choices we made in the method, the difficulty to reconstruct convective precipitation comes from the basic principle of downscaling methods. Convective precipitation are short events with huge amount of precipitation, and are not detectable at a large-scale resolution. This is why climate models are always increasing in resolution, to take into account the small case processes leading to convective precipitation. Even with a wind and humidity predictor in SANDHY, it is really difficult to relate a convective local event to a specific pattern in large scale predictors. This can be added in the data limitations.

*Somewhere, probably in the conclusions, the authors could / should include a sentence or two about application of the SCOPE approach, and specifically of the three 'improvements' imposed after the SANDHY step, to other settings. The authors have noted the positive aspects of highresolution time series available for the UK. Could they here make a comment about the opposite situation: larger areas with scarce or no data? Could they or anyone who wishes to reproduce their effort for other regions even imagine this work without the existence of Safran? How would a country of large area and broad range of land surfaces develop information and skills necessary to repeat a SCOPE-like reconstruction? Works nicely for France. Could it work elsewhere?*

As this is a data paper and not a methodological paper, the authors didn't want to emphasize the conclusion on the method (SCOPE), but preferred to present the dataset (SCOPE Climate) and his advantages. The last point was already discussed in the first comment, reconstructing data with SCOPE requires long observations. This would be an interesting work to apply SCOPE on point observations, knowing that the quality of the resulting dataset will depend on the quality of 20CR over the corresponding area, and on the quality of local observations.

**References**

Caillouet, L., Vidal, J.-P., Sauquet, E. and Gradd, B.: Probabilistic precipitation and temperature downscaling of the twentieth century reanalysis over France. Clim. Past, 12, 635–662, doi: 10.5194/cp-12-635-2016, 2016.

Vidal, J.-P., Martin, E., Franchistéguy, L., Baillon, M., and Soubeyroux, J.-M.: A 50-year high-resolution atmospheric reanalysis over France with the Safran system, Int. J. Climatol., 30, 1627– 1644, doi:10.1002/joc.2003, 2010.